# Beyond Memristors: Neuromorphic Computing Using Meminductors

**DOI:** 10.3390/mi14020486

**Published:** 2023-02-19

**Authors:** Frank Zhigang Wang

**Affiliations:** Division of Computing, Engineering & Mathematics, University of Kent, Canterbury CT2 7NZ, UK; f.z.wang@kent.ac.uk

**Keywords:** memristor, meminductor, novel computing architectures, non-Turing machine, neuromorphic computing, deep learning, brain-inspired computers

## Abstract

Resistors with memory (memristors), inductors with memory (meminductors) and capacitors with memory (memcapacitors) play different roles in novel computing architectures. We found that a coil with a magnetic core is an inductor with memory (meminductor) in terms of its inductance *L*(*q*) being a function of charge *q*. The history of the current passing through the coil is remembered by the magnetization inside the magnetic core. Such a meminductor can play a unique role (that cannot be played by a memristor) in neuromorphic computing, deep learning and brain-inspired computers since the time constant ($t_{0}=\sqrt{LC}$) of a neuromorphic *RLC* circuit is jointly determined by the inductance L and capacitance C, rather than the resistance R. As an experimental verification, this newly invented meminductor was used to reproduce the observed biological behavior of amoebae (the memorizing, timing and anticipating mechanisms). In conclusion, a beyond-memristor computing paradigm is theoretically sensible and experimentally practical.

## 1. Introduction

A memristor is an ideal candidate for non-Turing machines due to its compact processing-in-memory architecture. As a sister of the memristor (resistor-with-memory), inductor-with-memory (meminductor) has a unique role to play in neuromorphic computing systems, novel computing architectures and dynamical neural networks.

An inductor, typically consisting of an insulated wire wound into a coil, stores energy in a magnetic flux φ surrounding it when a current i flows through it. When the current changes, the time-varying magnetic flux induces a voltage across the coil, described by Faraday’s law [1]. Such an inductor is characterized by its inductance $L=\frac{\varphi }{i}$. In SI, the unit of inductance is the henry (*H*). As shown in Figure 1, by adding a magnetic core made of a ferromagnetic material, such as iron, inside the coil, the magnetizing flux from the coil induces magnetization in the material, increasing the magnetic flux. The high permeability of a ferromagnetic core can increase the inductance of a coil by a factor of several thousand over what it would be without it [1].

Organisms such as amoebae exhibit primitive learning and the memorizing, timing and anticipating mechanisms. Their adaptive behavior was emulated by a memristor-based *RLC* circuit [2]. Motivated by this work, we will design a meminductor-based neuromorphic architecture that self-adjusts its inherent resonant frequency in a natural way following the external stimuli frequency. In contrast to the previous work, our innovation is that this architecture uses a unique meminductor to increment its time constant and subsequently decrement its resonant frequency to match the stimuli frequency. It is our intention to use this architecture to help better investigate the cellular origins of primitive intelligence. This is also the significance of this sort of research in terms of not only understanding the primitive learning but also developing a novel computing architecture.

In this article, we first prove that a coil structure with a magnetic core is, in fact, a meminductor, since its inductance is no longer a constant, and then experimentally verify this new device in neuromorphic computing.

## 2. LLG Model for the Coil Core Structure

Next, we produce a theory to physically describe the current–flux interaction in a conducting coil with a magnetic core. For the sake of convenience, the magnetic core is assumed to be a single-domain cylinder with uniaxial anisotropy in the approximate sense: the magnetization is uniform and rotates in unison [3]. In an ideal case, there is a negligible amount of eddy current damping and parasitic “capacitor” effect.

It was found that the rotational process dominates the fast reversal of square loop ferrites with a switching coefficient $S_{w}=0.2\;Oe\cdot \mu s$ [4]. The rotational model for the coil core structure is shown in Figure 2.

The Landau–Lifshitz–Gilbert equation [5,6] is
$$\left(1+g^{2}\right)\frac{d\vec{M_{S}}\left(t\right)}{dt}=-\left|\gamma \right|\left[\vec{M_{S}}\left(t\right)\times \vec{H}\right]-\frac{g\left|\gamma \right|}{M_{S}}\left[\vec{M_{S}}\left(t\right)\times \left(\vec{M_{S}}\left(t\right)\times \vec{H}\right)\right]$$
where *g* is the damping factor and γ is the gyromagnetic ratio.

The first term of the right-hand side can be rewritten as $-\left|\gamma \right|\vec{M_{S}}\left(t\right)\times \vec{H}=-\left|\gamma \right|(M_{S}sin\theta sin\psi H\vec{i}-M_{S}sin\theta cos\psi H\vec{j})$. This term has no $\vec{k}$ component (along *Z*) and does not contribute to *M_Z_*. 

The second term can be rewritten as
$$-\frac{g\left|\gamma \right|}{M_{S}}\left[\vec{M_{S}}\left(t\right)\times \left(\vec{M_{S}}\left(t\right)\times \vec{H}\right)\right]$$
$$=-\frac{g\left|\gamma \right|}{M_{S}}(M_{S}sin\theta cos\psi \vec{i}+M_{S}sin\theta sin\psi \vec{j}$$
$$+M_{S}cos\theta \vec{k})\times \left[M_{S}sin\theta sin\psi H\vec{i}-M_{S}sin\theta cos\psi H\vec{j}\right]$$
$$=-\frac{g\left|\gamma \right|}{M_{S}}\left[-M_{S}sin\theta cos\psi M_{S}sin\theta cos\psi H-M_{S}sin\theta sin\psi M_{S}sin\theta sin\psi H\right]\vec{k}$$
$$=g\left|\gamma \right|M_{S}H\left[sin^{2}\theta cos^{2}\psi +sin^{2}\theta sin^{2}\psi \right]\vec{k}=g\left|\gamma \right|M_{S}Hsin^{2}\theta \vec{k}$$
$$=g\left|\gamma \right|M_{S}H(1-cos^{2}\theta )\vec{k}=g\left|\gamma \right|M_{S}H\left[1-{\left(\frac{M_{Z}}{M_{S}}\right)}^{2}\right]\vec{k}$$

From the above, we can obtain the following equation:(1)$$\left(1+g^{2}\right)\frac{dM_{Z}\left(t\right)}{dt}=g\left|\gamma \right|M_{S}H\left[1-{\left(\frac{M_{Z}}{M_{S}}\right)}^{2}\right]$$

Assuming $m\left(t\right)=\frac{M_{Z}\left(t\right)}{M_{S}}$, we can obtain
(2)$$\frac{dm\left(t\right)}{dt}=\frac{g\left|\gamma \right|H}{\left(1+g^{2}\right)}\left[1-m^{2}\left(t\right)\right]=\frac{1}{S_{W}}i\left(t\right)\left[1-m^{2}\left(t\right)\right]$$

The threshold for magnetization switching is automatically taken into account because the switching coefficient is defined based on the threshold field *H*_0_, which is one to two times the coercive force *H_C_* [3,7,8].

The hyperbolic function *tanh* has $\frac{d}{dx}\tanh x=1-tanh^{2}x$ and the derivative of a function of function has $\frac{du}{dx}=\frac{du}{dy}\frac{dy}{dx}$; therefore, it is reasonable to assume that
(3)$$m\left(t\right)=\tanh\left[\frac{q\left(t\right)}{S_{W}}+C\right],$$
where $\frac{d}{dt}q\left(t\right)=i\left(t\right)$ and *C* is a constant of integration, such that $C=tanh^{-1}m_{0}$ if *q*(*t =* 0) *=* 0 (assuming the charge does not accumulate at any point) and *m*_0_ is the initial value of *m*.

*dM_z_/dt* can be observed by the voltage *v*(*t*) induced:(4)$$\mu_{0}S\frac{dM_{z}}{dt}=S\frac{dB_{z}}{dt}=\frac{d\varphi_{z}}{dt}=-v\left(t\right)$$
where *μ*_0_ is the permeability and *S* is the cross-sectional area. 

Equation (4) results in
(5)$$\varphi =\mu_{0}SM+C^{\prime }=\mu_{0}SM_{S}m+C^{\prime }$$
where $C^{\prime }$ is another constant of integration.

Combining Equation (3) and Equation (5) and assuming φ(t=0)=0, we have $C^{\prime }=-\mu_{0}SM_{S}m_{0}$, so
(6)$$\varphi =\mu_{0}SM_{s}\left[\tanh\left(\frac{q}{S_{W}}+tanh^{-1}m_{0}\right)-m_{0}\right].$$

Beyond the first-order setting, a second-order circuit element, such as a meminductor, requires double-time integrals of voltage and current, namely, σ=∫qdt=∬idt and ρ=∫φdt=∬vdt. With the use of these additional variables [8,9], we accommodate a meminductor, a memcapacitor and other second-order circuit elements with memory. By integrating Equation (6), we have
(7)$$\rho =\int_{\tau =-\infty }^{t}\varphi d\tau =\mu_{0}SM_{s}\int_{\tau =-\infty }^{t}\left[\tanh\left(\frac{q}{S_{W}}+tanh^{-1}m_{0}\right)-m_{0}\right]d\tau .$$

Since ∫tanhxdx=ln(coshx)+C, we have
(8)$$\rho =\mu_{0}SM_{s}\;ln\left\{\cosh\left[\tanh\left(\frac{q}{S_{W}}+tanh^{-1}m_{0}\right)-m_{0}\right]\right\}+C≜\hat{\rho }\left(q\right).$$

Therefore, we have
(9)L=φi=μ0SMs[tanh(qSW+tanh−1m0)−m0]dqdt≜L(q)
where the denominator is still a function of the charge $q=\hat{q}\left(t\right)$ since $\frac{dq}{dt}=i\left(t\right)=i\left[{\hat{q}}^{-1}\left(q\right)\right]$.

Based on Equation (8), a typical ρ−q curve is depicted in Figure 3 with *m*_0_
*=* −0.964 (this value reflects the intrinsic fluctuation; otherwise, ***M*** reverts to the stable equilibria $m_{0}=\pm 1$).

## 3. Experimental Verification of the Rotational Model

To verify the validity/accuracy of the above rotational model, Equation (3) with H(t)∝i(t) is used to reproduce various *m–H* loops in Figure 4.

As a comparison, a typical *m–H* loop of real-world magnetic materials is displayed in Figure 5. The above simulations clearly validate Cushman’s conclusion that “the rotational model is applicable to the driving current of an arbitrary waveform” [3].

As another comparison, a simulated loop based on $m=\tan h\left(A*\left(H\pm H_{C}\right)\right)$] is displayed in Figure 6. The equivalence of formula $m=\tan h\left(A*\left(H\pm H_{C}\right)\right)$] and formula $m\left(t\right)=tanh\left[\frac{1}{S_{W}}\left(q\left(t\right)\pm S_{W}tanh^{-1}\left|m_{0}\right|\right)\right]$ indicates that the rotational model is good enough to reproduce a sine-wave response.

## 4. Simulations and Experiments of a Coil Core Meminductor for Neuromorphic Computing

Nature exhibits unconventional ways of processing information. Taking amoebae as an example, they display memorizing, timing and anticipating mechanisms, which may represent the origins of primitive learning. A circuit element with memory can be used to mimic these behaviors in terms of being plastic according to the dynamic history [13,14,15]. 

As shown in Figure 7, a simple *RLC* neuromorphic circuit using a coil core meminductor, *L*(*q*), is designed. The temperature controlling the motion of an amoeba is analogous to the input voltage, *V_in_*, whereas the output voltage, *V_out_*, is analogous to the locomotive speed of the amoeba.

With the progress of time, the circuit’s resonance frequency automatically scans the following frequency range:(10)$$f_{0}=\frac{1}{2\pi \sqrt{L\left(q\right)C}}=\frac{1}{2\pi \sqrt{L\left(\int i\left(t\right)dt\right)C}}$$

When the ramping circuit resonance frequency, *f*_0_, hits the (temperature) stimulus frequency, *f_sti_,* at a time point, a resonance is triggered. 

This neuromorphic circuit in Figure 7 using a coil core meminductor reasonably reproduces a behavior that was observed on amoebae: in response to the input stimulus pulses (representing the temperature drops), the circuit reduces the amplitude of its output (representing the amoeba’s speed) at the corresponding time points. As demonstrated in Figure 8, long-lasting responses for spontaneous in-phase slow down (SPS) [13,14] are both simulated and tested experimentally: the amoeba being exposed to the three temperature drops slows down or even stops at the corresponding time points *S*_1_, *S*_2_ and *S*_3_. Remarkably, the amoeba is found to slow down even if the temperature drops do not occur at *C*_1_, *C*_2_ and *C*_3_ (that are naturally anticipated by the amoeba after the three consecutive drops are experienced at *S*_1_, *S*_2_ and *S*_3_).

The experimental setup of the neuromorphic circuit in Figure 8 is as follows: L[q(t)]=L[∫i(t)dt] starts at 2 *H* and then decreases by 20% after each stimulus pulse. The circuit’s resonance frequency, determined by the staircased *L*(*q*) (Figure 3)*,* increases itself with the increased number of oncoming stimulus pulses. This simulation in Figure 8a agrees with our experiment in Figure 8b on a hardware emulator built with a dsPIC30F2011 microcontroller, an MCP4261 digital potentiometer and a differential 12-bit ADC converter [15].

This experiment vividly demonstrates amoebae’s three mechanisms: 1. the memorizing mechanism (the amoeba remembers the three temperature drops at *S*_1_, *S*_2_ and *S*_3_); 2. the timing mechanism (the amoeba slows down at the correct time points *C*_1_, *C*_2_ and *C*_3_ despite no temperature drops at these time points); and 3. the anticipating mechanism (the reason the amoeba slows down actively is because it anticipates the future possible drops at *C*_1_, *C*_2_ and *C*_3_ based on its memory of *S*_1_, *S*_2_ and *S*_3_ although these temperature drops at *C*_1_, *C*_2_ and *C*_3_ do not occur). Remarkably, these memorizing/timing/anticipating mechanisms are implemented by our newly invented coil core meminductor in terms of using the magnetization to remember the current history, adapting automatically the time constant determined by *L*(*q*) to the stimulus and triggering the resonance, respectively.

This neuromorphic circuit is a deep learning neural network [16] with multiple layers between the input and output layers, as shown in Figure 9. The meminductor *L*(*q*) and capacitor *C* store energy in the form of magnetic flux and electric field, respectively, whereas resistor *R* only consumes energy. Energy can be transferred from one form to the other, which is oscillatory with a resonance frequency ($f_{0}=\frac{1}{2\pi \sqrt{L\left(q\right)C}}$). The resistance *R* dampens the oscillation, diminishing it with time. Not strictly speaking, such a damped oscillation may be vividly approximated by $e^{-\alpha t}\sin2{\pi f}_{0}t$, where $\alpha =\frac{R}{2L\left(q\right)}$ is the damping factor.

## 5. Discussion and Conclusions

Memristors (resistors with memory), meminductors (inductors with memory) and memcapacitors (capacitors with memory) have different roles in neuromorphic computing systems, novel computing architectures and dynamical neural networks. In this study, we found that a coil with a magnetic core is, in fact, an inductor with memory (meminductor) in terms of its inductance being a function of the charge. This meminductor can play a unique role (that cannot be played by a memristor) in neuromorphic computing [17,18], deep learning [16] and brain-inspired computing [19,20,21] since the time constant ($t_{0}=\sqrt{LC}$) of a neuromorphic *RLC* circuit is jointly determined by the inductance L and capacitance C, rather than the resistance R. As an experimental verification, this new meminductor was used to reasonably reproduce the observed biological behavior of amoebae, in which the resonance frequency tracks the stimulus frequency. In conclusion, a beyond-memristor computing paradigm is theoretically sensible and experimentally practical.

Nature exhibits unconventional ways of storing and processing information, and circuit elements with memory mimic the dynamical behaviors of some biological systems in terms of being plastic according to the history of the systems. As a practical application, the Pavlovian experiment on conditioned reflex is reproduced by a memristor neural network with the aid of the so-called “delayed switching” effect [22,23]. In this application, the total length of the stimuli sequence, the frequency of the stimuli sequence and the spike width are carefully adjusted such that the time delay point of the memristor synapse should not be exceeded while only one neuron fires. In many applications, it is not feasible and possible to solve the problems with conventional computational models and methods (i.e., the Turing machine [24,25,26,27] and the von Neumann architecture [28,29,30,31]). As demonstrated above, neuromorphic architectures may help.

Understanding the brain with non-linear dynamics and extreme complexity is still a great challenge since the human brain has 10^11^ neurons and 10^14^ synapses (each neuron is connected to up to 20,000 synapses) [32,33,34,35,36]. By coincidence, as one of the simplest creatures or organisms existing on earth, unicellular amoebae display some mysterious brain-like behaviors in terms of controlling their actions [37,38,39,40,41]. Their memorizing, timing and anticipating mechanisms may represent the origins of primitive learning. 

The evolution of life includes the process of evolving intelligence in charge of controlling and predicting their behavior. In 1952, Hodgkin and Huxley developed an equivalent circuit to explain the initiation/propagation of action potentials and the underlying ionic mechanisms in the squid giant axon [17,42,43,44,45,46]. They were awarded the Nobel Prize in Physiology or Medicine for this work in 1963. In the so-called Hodgkin–Huxley model, an electrical circuit representing each cell consists of a linear resistor, a capacitor, three batteries, and two unconventional elements identified by Hodgkin and Huxley as time-varying resistors. In 2012, these two potassium and sodium time-varying resistors were substituted by a potassium ion-channel memristor, and a sodium ion-channel memristor, respectively [18,19]. This presents great progress in neural physiology and brain science in over 70 years in terms of exploring the origins of primitive learning from an evolutionary perspective.

In this work, we developed a meminductor-based neuromorphic architecture that self-adjusts its inherent resonant frequency in a natural way following the external stimuli frequency. In contrast to the previous work, our innovation is that this architecture uses a unique meminductor to increment its time constant and subsequently decrement its resonant frequency to match the stimuli frequency. This architecture may help better investigate the cellular origins of primitive intelligence [47,48,49]. This sort of research is significant in terms of not only understanding the primitive learning but also developing a novel computing architecture, which will be much more integrated with our physical and social environment, capable of self-learning, as well as processing and distributing big data at an unprecedented scale [50,51]. This will require new designs, new theories, new paradigms and close interactions with application experts in the sense that new bio-inspired (neurosynaptic) and non-Turing-inspired computing platforms are moving away from traditional computer architecture design [51].

## Figures and Tables

**Figure 1 micromachines-14-00486-f001:**
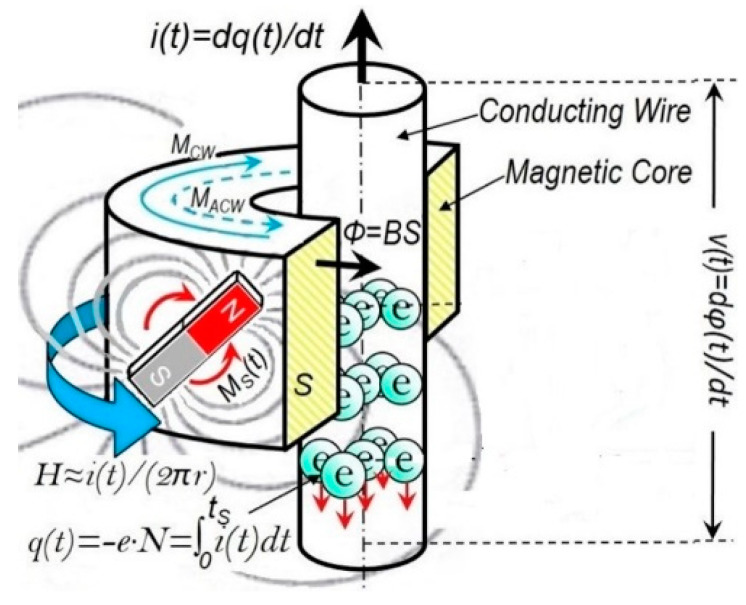
In this study, we found that a coil with a magnetic core is, in fact, an inductor with memory (meminductor) in terms of its inductance being a function of the charge. The Oersted field generated by the current *i* rotates or switches the magnetization *M* inside the core and consequently the switched flux *φ* induces a voltage v across the conductor. The history of the current passing through the coil [∫i(t)dt=q(t)] is remembered by the magnetization inside the magnetic core.

**Figure 2 micromachines-14-00486-f002:**
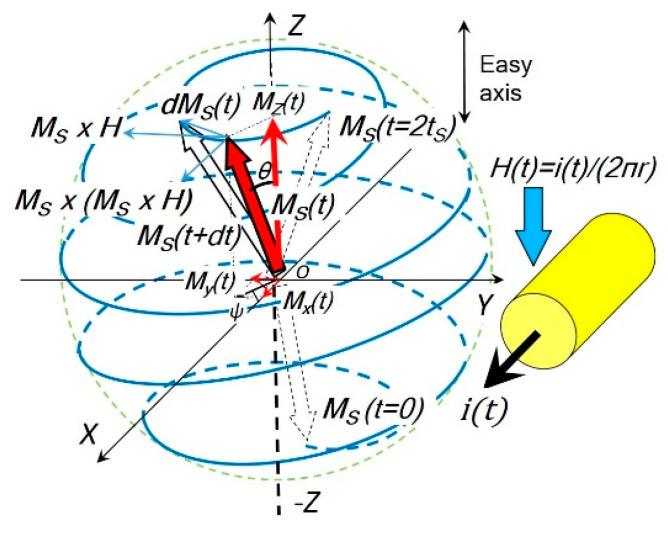
The rotational model used in the coil core structure. If the magnetic field ***H*** is applied in the *Z* direction, the saturation magnetization vector ***M_S_***(*t*) follows a precession trajectory (blue) from its initial position (*θ_0_* ≈ *π, m_0_* ≈ −1) and the angle *θ* decreases with time continuously until (*θ* ≈ *0, m* ≈ 1), i.e., the magnetization ***M_S_***(*t*) reverses itself and is eventually aligned with the magnetic field ***H***.

**Figure 3 micromachines-14-00486-f003:**
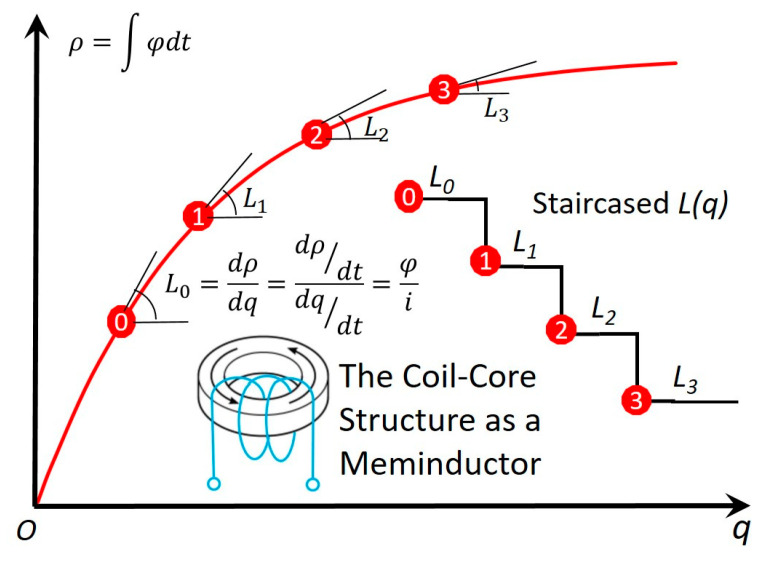
The constitutional *ρ–q* curve of the meminductor. It complies with the three criteria for the ideality of an ideal circuit element with memory [10,11]: a. nonlinear; b. continuously differentiable; and c. strictly monotonically increasing. With the accumulation of the charge, $L\left(q\right)=\frac{d\rho }{dq}$ decreases like a staircase.

**Figure 4 micromachines-14-00486-f004:**
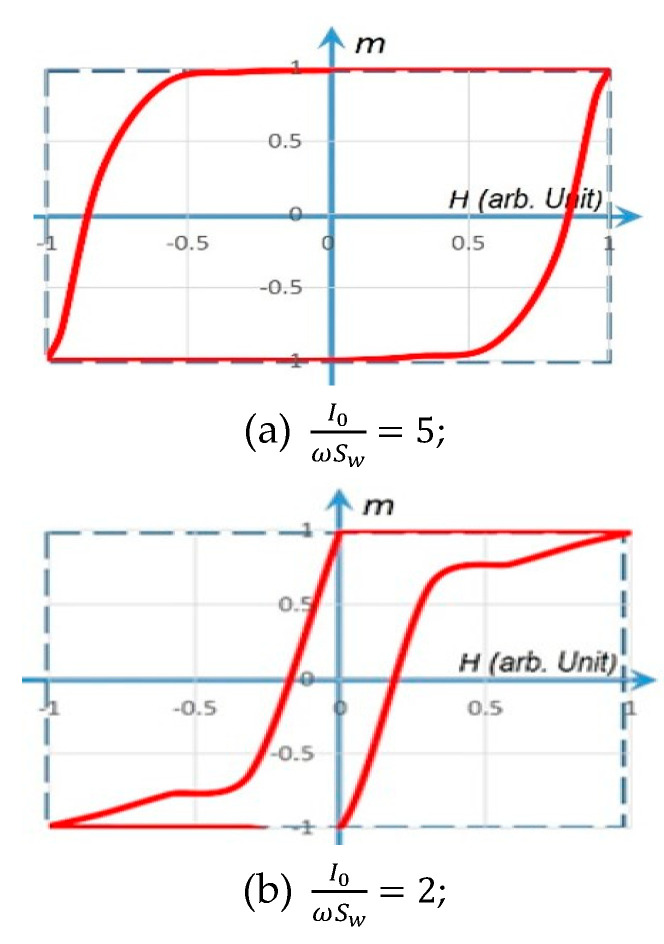
The *m–H* hysteresis loops simulated by the rotational model. The solid line in red represents a gradual *m*(*t*) rotation (with a finite slope) under $H\left(t\right)\propto i\left(t\right)=I_{0}\sin\omega t$, $m_{0}=\pm 0.99$. The dashed line in blue represents a fast *m*(*t*) rotation (with an infinite slope) under a step-function *H*.

**Figure 5 micromachines-14-00486-f005:**
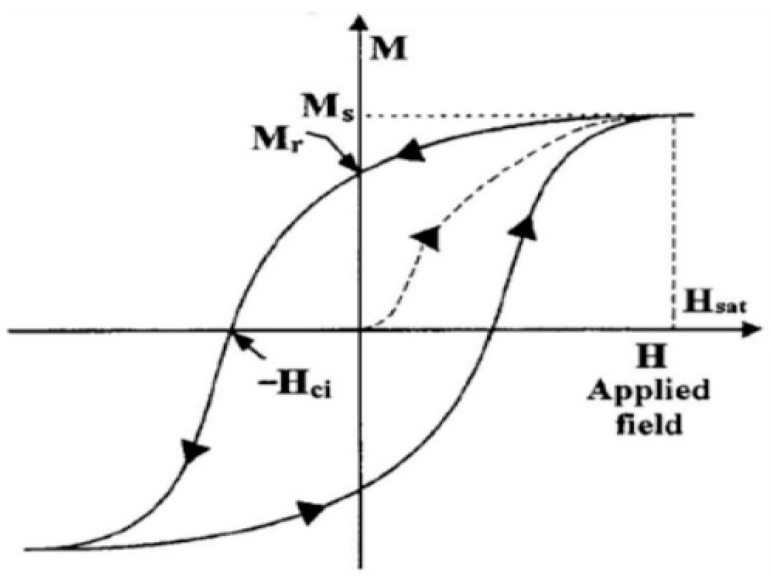
A typical *m–H* loop of real-world magnetic materials [12].

**Figure 6 micromachines-14-00486-f006:**
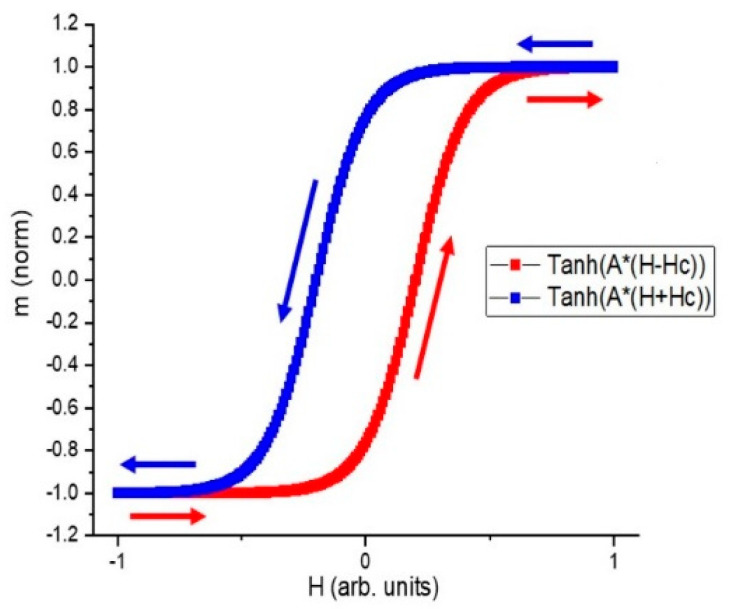
The simulated *m–H* loop based on $m=\tan h\left(A*\left(H\pm H_{C}\right)\right)$] (*H_C_* is the coercive force) with a sine-wave input current. Two *tanh* values are used, and a horizontal shift is applied to each branch to obtain hysteresis.

**Figure 7 micromachines-14-00486-f007:**
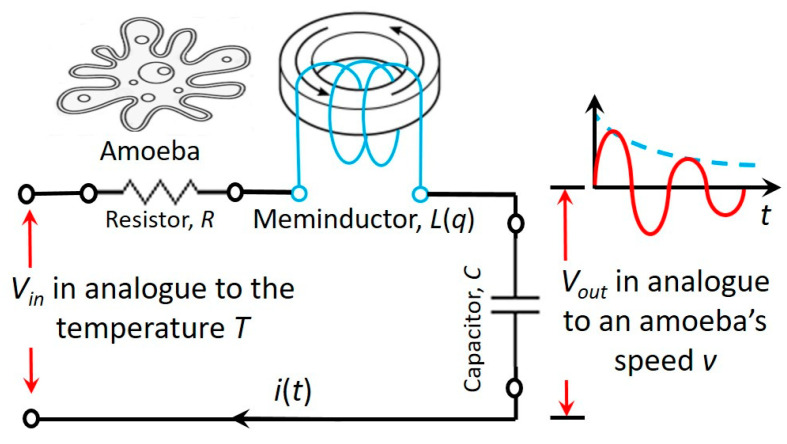
An *RLC* neuromorphic circuit using a coil core meminductor, *L*(*q*), to scan a frequency range. An amoeba’s behavior is simulated with the damped oscillations of this circuit.

**Figure 8 micromachines-14-00486-f008:**
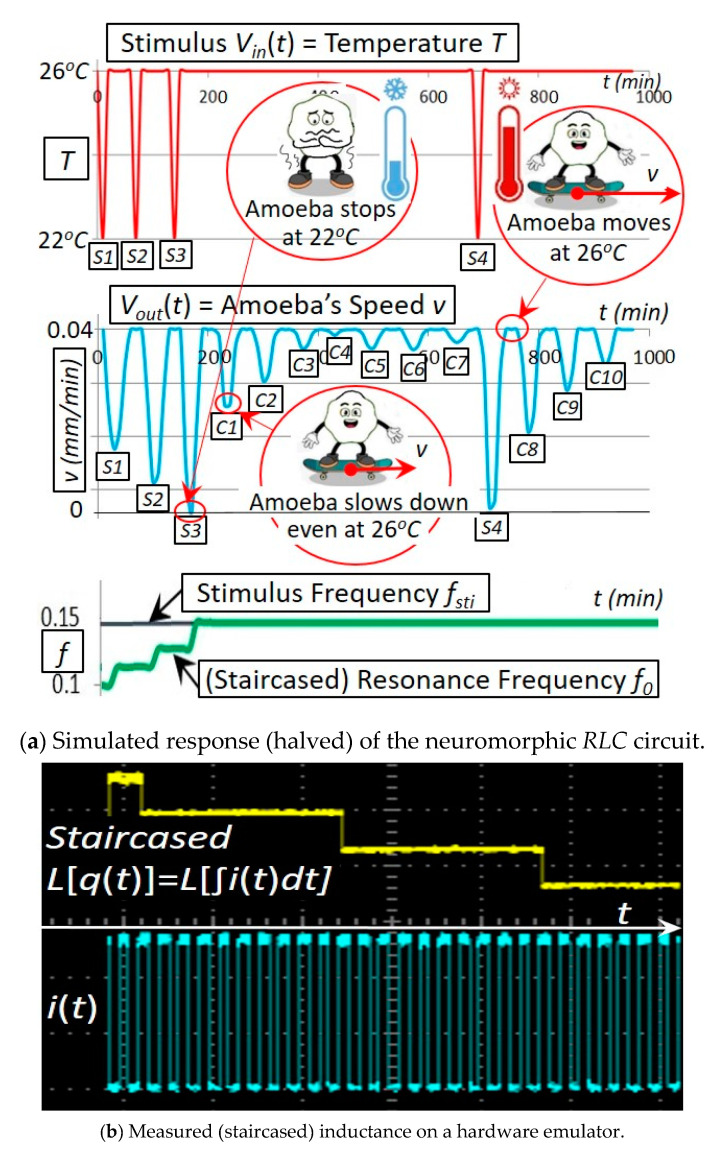
Simulated and experimental responses of the neuromorphic circuit. L[q(t)]=L[∫i(t)dt] starts at 2 *H* and then decreases by 20% after each stimulus pulse. The circuit’s resonance frequency, determined by the staircased *L*(*q*) (Figure 3)*,* increases itself with the increased number of oncoming stimulus pulses. This simulation in (**a**) agrees with our experiment in (**b**) on a hardware emulator built with a dsPIC30F2011 microcontroller, an MCP4261 digital potentiometer and a differential 12-bit ADC converter.

**Figure 9 micromachines-14-00486-f009:**
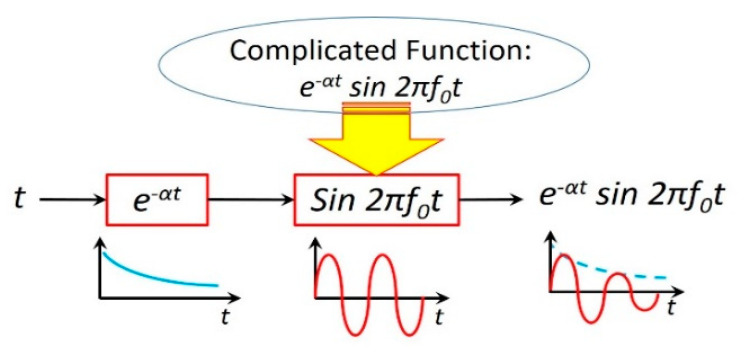
The neuromorphic *RLC* circuit in Figure 7 is a deep learning neural network with multiple layers between the input and output layers. The complicated function $e^{-\alpha t}\sin2{\pi f}_{0}t$ is decomposed into two simple functions: $e^{-\alpha t}$ and $\sin2{\pi f}_{0}t$, each of which can be implemented in one layer. The former is determined by *R* and *L*(*q*)*,* whereas the latter is determined by *L*(*q*) and *C*.

## Data Availability

Most of the data generated and analysed during this study are included in this published article. The additional data are available on request from the corresponding author.

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
