# Peer review of "Beyond Memristors: Neuromorphic Computing Using Meminductors"

_micromachines, 2023, doi:10.3390/mi14020486_

Round 1
Reviewer 1 Report
1. The abstract of the article is incomplete, and the main work and conclusions should be included in itï¼›
2. The introduction is written too simple, it is suggested to supplement the background of selecting the topic, the research intention and significance, method and innovation, etcï¼›
3. The memristor-based circuits to mimic the biological behaviour of amoebae have been extensively studied. Compared with these circuits, what are the advantages of the proposed meminductor-based amoebae circuit.
Reviewer 2 Report
I find the paper interesting, since it proposes a new idea of using inductors as meminductor devices. I think that the presentation of the paper should be improved a little bit, specifically:
- Figure(s) 4 are too big and the resolution is too low, the figure can be improved by reducing the size of each subfigure and optimizing their placement.
- The explanation of the experimental setup is only presented in the caption of the figure, and it should be reported also in the main text of the paper.
- While I like the idea and the case of study, I fail to understand how the meminductor can be used for practical applications. I believe that the paper can be improved by adding a short paragraph where possible practical applications are disclosed, this will improve the general readability of the paper for people that are not experts.
Round 2
Reviewer 1 Report
The paper has been well revised according to the reviewer‘s comments and I think it can be accepted in current version.